# AoSsk1, a Response Regulator Required for Mycelial Growth and Development, Stress Responses, Trap Formation, and the Secondary Metabolism in *Arthrobotrys oligospora*

**DOI:** 10.3390/jof8030260

**Published:** 2022-03-03

**Authors:** Ke-Xin Jiang, Qian-Qian Liu, Na Bai, Mei-Chen Zhu, Ke-Qin Zhang, Jin-Kui Yang

**Affiliations:** State Key Laboratory for Conservation and Utilization of Bio-Resources, Key Laboratory for Microbial Resources of the Ministry of Education, School of Life Sciences, Yunnan University, Kunming 650091, China; kxjiang@mail.ynu.edu.cn (K.-X.J.); liuqianqian614@163.com (Q.-Q.L.); baina@mail.ynu.edu.cn (N.B.); zmc201789@163.com (M.-C.Z.); kqzhang1@ynu.edu.cn (K.-Q.Z.)

**Keywords:** *Arthrobotrys oligospora*, response regulator Ssk1, asexual development, trap formation, secondary metabolism, pathogenicity

## Abstract

Ssk1, a response regulator of the two-component signaling system, plays an important role in the cellular response to hyperosmotic stress in fungi. Herein, an ortholog of *ssk1* (*Ao**ssk1*) was characterized in the nematode-trapping fungus *Arthrobotrys oligospora* using gene disruption and multi-phenotypic comparison. The deletion of *Aossk1* resulted in defective growth, deformed and swollen hyphal cells, an increased hyphal septum, and a shrunken nucleus. Compared to the wild-type (WT) strain, the number of autophagosomes and lipid droplets in the hyphal cells of the Δ*Aossk1* mutant decreased, whereas their volumes considerably increased. *Aossk1* disruption caused a 95% reduction in conidial yield and remarkable defects in tolerance to osmotic and oxidative stress. Meanwhile, the transcript levels of several sporulation-related genes were significantly decreased in the Δ*Aossk1* mutant compared to the WT strain, including *abaA*, *brlA*, *flbC*, *fluG*, and *rodA*. Moreover, the loss of *Aossk1* resulted in a remarkable increase in trap formation and predation efficiency. In addition, many metabolites were markedly downregulated in the Δ*Aossk1* mutant compared to the WT strain. Our results highlight that AoSsk1 is a crucial regulator of asexual development, stress responses, the secondary metabolism, and pathogenicity, and can be useful in probing the regulatory mechanism underlying the trap formation and lifestyle switching of nematode-trapping fungi.

## 1. Introduction

The transmission of extracellular signals to intracellular targets is mediated by a network of interacting proteins that regulate a large number of cellular processes in eukaryotic cells. Intracellular signal transduction systems regulate growth, development, and reproduction [1]. The mitogen-activated protein kinase (MAPK) signaling cascade is the most conserved signal module in all eukaryotes and consists of three layers of protein kinases: an MAP kinase, an MAP kinase (MAPKK), and an MAPKK kinase (MAPKKK) [2]. Six types of MAPKs have been identified in the model yeasts *Saccharomyces cerevisiae* and *Candida albicans* [3,4], and three types of MAPKs (Slt2, Hog1, and Fus3) have been identified in plant pathogenic and entomopathogenic fungi [5,6]. The high osmotic pressure glycerol (Hog) pathway is essential for the survival of yeast and filamentous fungi in a high osmotic environment [5,7]. In *S. cerevisiae*, the Hog1 pathway mainly consists of the Ssk2/Ssk22–Pbs2–Hog1 cascade [8]. Hog1 has many effects on osmotic stress, such as the regulation of the expression of various genes, protein translation, cell cycle progression, and the synthesis of glycerol to respond to changes in the external environment [8,9].

Ssk1 is a typical two-domain response regulator protein in the two-component signal transduction system of the Hog1 pathway. Under normal conditions, Ssk1 is phosphorylated. When hypertonic pressure is detected, Ssk1 rapidly dephosphorylates and activates the Hog1 pathway, which ultimately controls the concentration of intracellular glycerol under hypertonic stress [10,11]. Ssk1 has been identified in several yeast species. For example, Ssk1 is essential for the mycelial growth and virulence of *C. albicans* [12], and the Δ*ssk1* mutant is sensitive to several oxidative stress agents [13]. Ssk1 is vital for fungal pathogenesis, including *Candida* antifungal drug resistance and determination of the cell wall structure in *Candida auris* [14]. The role of Ssk1 has been identified in several filamentous species, including *Beauveria bassiana* [15], *Alternaria alternata* [16], and *Verticillium dahliae* [17]. In *B. bassiana*, the disruption of *Bbssk1* causes a reduction in conidial yield and virulence, as well as a significant decrease in tolerance to chemical stressors [15]. Similarly, VdSsk1 is involved in the stress response, melanin biosynthesis, and maximum virulence of *V. dahliae* [17]. Therefore, Ssk1 orthologs play a vital role in the growth, stress response, and pathogenicity of yeasts and pathogenic fungi.

Nematode-trapping (NT) fungi are a group of filamentous species that can produce special mycelial traps to capture and digest nematodes, and are found in all ecosystems worldwide [18]. They can form various traps, such as adhesive knobs, adhesive networks, and constricting rings, for nematode predation [19]. Trap formation is indispensable for NT fungi to capture nematodes, and is an indicator of their transition from a saprophytic to a predatory lifestyle [20,21]. *Arthrobotrys oligospora* is one of the best-studied NT fungi, and can produce an ingenious three-dimensional net (adhesive network) for nematode predation [22]. The genome of *A. oligospora* was sequenced in 2011, and an increasing number of studies have focused on the regulation of mycelial development and trap formation in this fungus. Several signaling proteins have been found to regulate trap formation and related phenotypic traits, including the regulators of G protein signaling [23,24] and MAPK cascades [25,26]. Recently, the membrane mucin Msb2 and Hog1 were identified in *A. oligospora*, which play an important role in osmosensing, trap morphogenesis, and conidiation [25]. To further probe the role of the two-component signaling system and Hog1 signaling in NT fungi, we characterized an orthologous Ssk1, a specific activator of MAPKKK functioning upstream of Hog1 in *A. oligospora*, by gene disruption, multi-phenotypic analysis, and metabolomic approaches. Studies on the two-component signaling system and Hog1 signaling in NT fungi may help to understand the mechanism of NT fungi responding to adverse environments, and may contribute to the application of NT fungi.

## 2. Materials and Methods

### 2.1. Strains and Culture Conditions

The fungus *A. oligospora* (ATCC24927) and the derived mutant strains were routinely inoculated on potato dextrose agar (PDA) medium at 28 °C, and these strains were preserved in the Microbial Library of the Germplasm Bank of Wild Species from Southwest China. The *Escherichia coli* strain DH5α (Takara, Shiga, Japan) was used as a recipient of plasmids pRS426 (cloning vector) and pCSN44 (containing hygromycin resistance gene), and *S. cerevisiae* (FY834) was used to screen the correctly recombined constructs on SC-Ura medium [27]. PDA, tryptone glucose (TG), and tryptone yeast extract glucose agar (TYGA) media were prepared and used for the phenotypic analysis of the fungal strains, as described previously [28]. *Caenorhabditis elegans* (strain N2) was cultivated in an oat medium at 26 °C for the induction of trap formation of the NT fungi.

### 2.2. Sequence Analysis of AoSsk1

The orthologous Ssk1 (AoSsk1, AOL_s00112g14) was searched in the *A. oligospora* genome using BlastP in the NCBI database, based on the orthologous Ssk1 in the model fungi *Aspergillus nidulans* (XP_680966) and *Neurospora crassa* (XP_011392892). The orthologs of Ssk1 from different fungi (Appendix A) were retrieved and downloaded from GenBank. The molecular mass and isoelectric point of AoSsk1 were analyzed using the online software pI/Mw (accessed 19 October 2020), and the similarity between AoSsk1 and other orthologs was analyzed using DNAman (version 5.22). A neighbor-joining tree was constructed using Mega (version 7.0) [29].

### 2.3. Disruption of Aossk1 and Its Verification

*Aossk1* was disrupted by homologous recombination, as described previously [27,30]. Briefly, the upstream and downstream homologous fragments of the *Aossk1* gene were amplified from *A. oligospora*, and the hygromycin fragment *hph* was used as a selection marker conferring hygromycin B resistance on transformants, amplified from the pCSN44 plasmid using paired primers (Appendix A). The pRS426 plasmid was digested with *Eco*RI and *Xho*I, and then it and the amplified fragments were co-transformed into *S. cerevisiae* (FY834) by electroporation. The recombinant plasmid pRS426–AoSsk1–hph was then isolated from the yeast. The gene fragment for *Aossk1* disruption was amplified from the recombinant plasmid pRS426–AoSsk1–hph using primers Aossk1–5f/Aossk1–3r (Appendix A), and was transformed into *A. oligospora* using the protoplast transformation method as described previously [31]. The putative transformants were selected on PDAS medium containing 200 µg/mL of hygromycin B (Amresco, Solon, OH, USA) [30] and further verified by PCR and Southern blotting, as described previously [32].

### 2.4. Analysis of Mycelial Growth, Colony Morphology, Conidiation, and Cell Nucleus

The *A. oligospora* wild-type (WT) and Δ*Aossk1* mutant strains were inoculated on PDA medium at 28 °C for seven days. To observe the mycelial growth and colony morphology, a 7 mm block from each WT and mutant strain was inoculated on PDA, TG, and TYGA plates at 28 °C for six days, and the colony diameters of WT and mutants were measured at 24 h intervals. To observe the hyphal cells, a glass slide was inserted into a PDA plate, and the fungal strains were incubated on PDA plates at 28 °C for seven days. Mycelial samples collected from the glass slide were stained with 20 μg/mL of calcofluor white (CFW; Sigma Aldrich, St. Louis, MO, USA), and the hyphal morphology was observed under a fluorescence microscopy (Leica, Mannheim, Germany) [33]. The mycelia were stained with 20 μg/mL of 4′,6-diamino-2-phenylindole (DAPI; Sigma-Aldrich, St. Louis, MI, USA) for 30 min, followed by three washes with phosphate buffer (pH 6.8–7.2), and then further stained with 20 μg/mL of CFW for 5 min. In addition, the mycelial samples were observed by scanning electron microscopy (SEM; Quanta-200, FEI, Hillsboro, OR, USA), as described previously [34,35].

The fungal strains were inoculated on CMY medium at 28 °C for 14 days. Then, the spores were washed with glass beads and filtered through a funnel, and spore yields were determined as described previously [26]. To determine the spore germination rate, 2 × 10^4^ spores were spread on a water agar (WA) plate, and the germinated spores were counted at 4, 8, and 12 h. The spore morphology was observed by SEM after inoculation of the strains on PDA plates at 28 °C for seven days. To study the nuclei, the fungal spores were stained with 20 μg/mL of DAPI for 30 min and observed by fluorescence microscopy [36].

### 2.5. Comparison of Stress Resistance

The fungal strains were inoculated into TG media supplemented with different concentrations of chemical stressors at 28 °C for six days. The chemical stressors employed for the experiment were as follows: Sorbitol (0.25, 0.5, and 0.75 M) and NaCl (0.1, 0.2, and 0.3 M) as osmotic stressors, SDS (0.01%, 0.02%, and 0.03%) and Congo red (30, 60, and 90 μg/mL) as cell wall-perturbing agents, and H_2_O_2_ (5, 10, and 15 mM) and menadione (0.05, 0.07, and 0.09 mM) as oxidative stressors. The relative growth inhibition (RGI) values of the strains were calculated as previously described [26].

### 2.6. Analysis of Autophagy, Lipid Droplets (LDs), and Endocytosis

To study autophagy in the WT and Δ*Aossk1* mutant strains, they were incubated on PDA plates with sterile coverslips at 28 °C for five days. Subsequently, the mycelia of the WT and mutant strains were treated with 30–50 µL of 100 mg/mL of monodansylcadaverine (MDC) staining solution at 37 °C in the dark for 30–40 min, and the images were observed by fluorescence microscopy. For the staining of LDs, a glass slide was inserted into a PDA plate, and the fungal strains were incubated on PDA plates at 28 °C for seven days. Hyphal samples were stained with 10 µg/mL of boron dipyrromethene (BODIPY) dye (Sigma-Aldrich, St. Louis, MO, USA) for 30 min in the dark, and the LDs were observed by fluorescence microscopy [37]. The mycelia were examined using transmission electron microscopy (TEM; Hitachi, Tokyo, Japan).

FM4-64 (Biotium, Californiaw, CA, USA) is a lipophilic styrenic cell dye that can be used for studying endocytosis [38]. The WT and Δ*Aossk1* mutant strains were cultured on PDA plates with sterile coverslips at 28 °C for five days, FM4-64 was diluted 40 times with double-distilled water for mycelial staining, and the changes in FM4-64 at 0, 5, and 10 min after entry into the hyphae were recorded through fluorescence microscopy.

### 2.7. Analysis of Trap Formation and Nematode Predation

A total of 2 × 10^4^ spores of the WT and Δ*Aossk1* mutant strains were evenly spread on a WA plate and incubated at 28 °C for three to four days. Then, ~400 nematodes (*C. elegans*) were added to the center of each plate. The traps and captured nematodes were counted under light microscopy (Olympus, Tokyo, Japan) at 12 h intervals. In addition, electron-dense bodies (EDs) in the trap cells of the WT and Δ*Aossk1* mutant strains were observed using TEM, as previously described [34].

### 2.8. Reverse Transcription Quantitative PCR (RT-qPCR)

To determine the expression of the genes associated with sporulation and fatty acid oxidation, the fungal strains were incubated on cellophane on PDA plates at 28 °C for three, five, and seven days. The mycelia were scraped from cellophane, and the total RNA from mycelia was isolated with an RNA Extraction Kit (Axygen, Jiangsu, China) and reverse transcribed to cDNA with a FastQuant RT Kit (with gDNA, Takara, Shiga, Japan). The cDNA from each strain was used as a template for RT-qPCR, and the *A. oligospora* β-tubulin gene (*tub*) was used as an internal standard. The specific primers for the target genes (Appendix A) were designed using the online software Primer3 (v. 0.4.0) (accessed 13 June 2021). RT-qPCR was performed as previously described [31]. The transcription levels of the target genes were calculated using the 2^−ΔΔCt^ method; briefly, fold changes were calculated using the following formula: 2^−ΔΔCt^, where ΔΔCt is ΔCt (treatment) −ΔCt (control), ΔCt is Ct (target gene) −Ct (*tub*), and Ct is the threshold cycle [39]. The relative transcription level (RTL) of the target genes was calculated as the ratio of the transcription level in the Δ*Aossk1* mutant to that in the WT cells.

### 2.9. Metabolomics Analysis of Aossk1

The WT and Δ*Aossk1* mutant strains were inoculated into 250 mL of PD broth at 28 °C (180 rpm) for six days; then, the fermentation broth and mycelia in the culture were separated and the hyphae were dried and weighed. Ethyl acetate (250 mL) was added to extract the fermentation broth, and ultrasonic extraction was performed for 40 min. The fermentation broth was allowed to stand for 12 h, and the crude extract was concentrated in ethyl acetate under vacuum. The crude extract was dissolved in 500 µL of analytic-grade methanol, and the solution was filtered through a 0.22 μm membrane filter. Triplicates of each group were used for subsequent liquid chromatography–mass spectrometry (LC–MS) using the Thermo Fisher Scientific Dionex Ultimate 3000 UHPLC system and a Thermo Fisher high-resolution Q precision focusing mass spectrometer (Thermo Fisher, Bremen, Germany), as described previously [23,40]. Compound Discoverer 3.0 software (Thermo Fisher Scientific, Waltham, MA, USA) was used for non-targeted metabolomics analysis.

### 2.10. Statistical Analysis

Data are expressed as standard deviations (SDs) of the means from three biological replicates. Prism 8.0 (GraphPad Software, San Diego, CA, USA) was used for the statistical analysis, and the variance between the WT and mutant strains was ranked by Tukey’s honestly significant difference (HSD) test. Statistical significance was set at *p* < 0.05.

## 3. Results

### 3.1. Analysis of the AoSsk1 Sequence

*Aossk1* contains an intron (68 bp), which encodes a 1077-amino-acid-long polypeptide with a predicted molecular mass of 114 kDa and an isoelectric point of 9.49. AoSsk1 contains a conserved receiver domain of the signal transduction response regulator (IPR001789). AoSsk1 shares a 57.3–90.3% sequence similarity with orthologs from other NT fungi (Appendix A), a 30.0–50.1% sequence similarity with orthologs from other filamentous fungi, and only a 22.5% sequence similarity with orthologs from *S. cerevisiae*. The protein sequences of orthologous Ssk1 from 17 different fungi were divided into two branches. The Ssk1 orthologs from six NT fungi were clustered into clade A, and the Ssk1 orthologs from other filamentous fungi were clustered into clade B (Appendix A).

### 3.2. Verification of Positive Transformants

The *Aossk1* gene was disrupted by replacing the target sequence with the *hph* fragment, and the positive transformants were screened on PDAS medium containing hygromycin. Genomic DNA was isolated from WT strains and transformants, and the primers yz-5f and yz-3r were used to verify positive transformants by PCR amplification (Appendix A). The sizes of the fragments from the WT cells and transformants were 2288 bp and 1649 bp, respectively. Then, the genomic DNA of the WT cells and transformants was digested with the restriction enzyme *Age*I for Southern blot analysis. A single band was observed, indicating that non-specific recombination did not occur (Appendix A). Finally, seven positive transformants were obtained, and three independent mutants (Δ*Aossk1-47*, Δ*Aossk1-93*, and Δ*Aossk1-94*) were used for further experiments.

### 3.3. Aossk1 Plays a Vital Role in Mycelial Growth, Morphology, and Hyphal Cell Nuclei

When the WT and Δ*Aossk1* mutant strains were incubated on PDA, TYGA, and TG plates at 28 °C for six days, the mycelial growth of the Δ*Aossk1* mutant was found to be remarkably slower than that of the WT strain (Figure 1A,B). For example, the colony diameter of the WT strain was 1.7 times that of the Δ*Aossk1* mutant on PDA medium, and 1.4 times that of the Δ*Aossk1* mutant on TYGA medium (Figure 1B). Additionally, more hyphal septa were observed in the mycelia of the Δ*Aossk1* mutant than in those of the WT strain using CFW staining (Appendix A), and the hyphae of the Δ*Aossk1* strain were significantly enlarged and swollen (Figure 1C and Appendix A). Meanwhile, the hyphal cells of the Δ*Aossk1* mutant were shorter than those of the WT strain (Figure 1D and Appendix A). Moreover, DAPI staining showed that the number of cell nuclei in the WT strain was between 6 and 22, while that in the Δ*Aossk1* mutant was only one to four (Figure 1E and Appendix A).

### 3.4. AoSsk1 Regulates LD Accumulation

The LDs in the mycelia of the WT and Δ*Aossk1* strains were observed using a BODIPY staining assay, and while numerous LDs were observed in the WT strain, the number of LDs decreased and the volume of partial LDs was remarkably increased in the Δ*Aossk1* mutant (Figure 2A). Large LDs were also observed in the hyphal cells of the Δ*Aossk1* mutant by TEM (Figure 2B). To probe the regulation of the fatty acid metabolism by AoSsk1, seven proteins involved in fatty acid oxidation were selected, namely, 3-oxoacyl-[acyl-carrier protein] reductase (AOL_s00004g288), phosphatidylinositol transporter (AOL_s00081g51), 3-ketoacyl-CoA ketothiolase (Kat1, AOL_s00210g122), 3-hydroxybutyryl-CoA dehydrogenase (AOL_s00110g113), acyl-CoA dehydrogenase (AOL_s00079g276), peroxisomal multifunctional beta-oxidation protein (Mfp, AOL_s00054g29), and peroxisomal ABC transporter (Pxa1, AOL_s00004g606), and their expression was compared between the Δ*Aossk1* mutant and WT strain. At three and seven days, the expression of all genes except *AOL_s00004g288* was significantly downregulated (*p* < 0.05) in the Δ*Aossk1* strain compared to the WT strain. At five days, the expression of four genes (*AOL_s00004g288*, *AOL_s00081g51*, *AOL_s00110g113*, and *AOL_s00079g276*) was remarkably upregulated in the Δ*Aossk1* mutant compared to the WT strain, and that of Mfp was downregulated in the Δ*Aossk1* strain (Figure 2C).

### 3.5. AoSsk1 Regulates Autophagy and Endocytosis

Upon staining with MDC dye, autophagosomes were visible in the WT and Δ*Aossk1* mutant strains. While the WT mycelia had numerous autophagosomes (Figure 3A) distributed in hyphal cells in a punctate pattern, there was a decrease in the number of autophagosomes in the hyphal cells of the Δ*Aossk1* mutant, as well as a remarkable increase in their volume (Figure 3A).

To test the effect of AoSsk1 on endocytosis in *A. oligospora*, we stained the WT and mutant strains with FM4-64 dye and observed endocytosis at three time points. While endocytosis was obvious in the WT strain at 5 min, that in the Δ*Aossk1* mutant was not (Figure 3B). In addition, more autophagic vacuoles were observed in the WT strain than in the Δ*Aossk1* mutant by TEM (Figure 3C).

### 3.6. Aossk1 Plays a Crucial Role in Sporulation, Spore Germination, and Cell Nuclei

The spore morphology observed by SEM showed an increase in the length of the spores and a decrease in the number of spores on the conidiophores in the Δ*Aossk1* mutant (Figure 4A). The deletion of the *Aossk1* gene caused a significant reduction in spore yields; while the WT strain produced 2.49 × 10^5^ spores cm^−2^ after being cultured on CMY medium for 14 days, the Δ*Aossk1* mutants only produced 1.26 × 10^4^ spores cm^−2^ (Figure 4B). Meanwhile, the spore germination rate of the Δ*Aossk1* mutant was remarkably lower than that of the WT strain at 4, 8, and 12 h post-incubation (hpi) (Figure 4C). To probe the effect of *Aossk1* on conidiation, seven sporulation genes, namely, *abaA*, *brlA*, *flbC*, *fluG*, *rodA*, *velB*, and *wetA*, were selected, and their transcript levels were compared between the Δ*Aossk1* mutant and WT strains at different growth stages. In the early and later stages of sporulation (three and seven days, respectively), the transcript levels of all genes except *wetA* were significantly downregulated (*p* < 0.05) in the Δ*Aossk1* mutant; meanwhile, in the middle stage of sporulation (five days), the transcript levels of all genes except *wetA* and *velB* were significantly downregulated (*p* < 0.05) in the Δ*Aossk1* mutant (Figure 4D). In addition, DAPI staining showed 12–18 nuclei per spore in the WT strain, and only four to eight nuclei per spore in the Δ*Aossk1* mutant (Figure 4E,F).

### 3.7. AoSsk1 Regulates Multiple Stress Responses

Next, we explored the effect of deleting the *Aossk1* gene on multiple stresses, including oxidative stress, osmotic stress, and resistance to cell-wall-perturbing agents, in *A. oligospora*. The growth of the Δ*Aossk1* mutants was strongly inhibited compared with that of the WT strain on the medium supplemented with osmotic stressors NaCl (0.1, 0.2, and 0.3 M) and sorbitol (0.25, 0.5, and 0.75 mM), and the mutants could not grow in the presence of 0.2 or 0.3 M NaCl and 0.5 or 0.75 mM sorbitol (Figure 5A). As a result, the RGI values of the Δ*Aossk1* mutant were significantly higher than those of the WT strain under the osmotic stresses of NaCl and sorbitol (Figure 5B). Similarly, the growth of Δ*Aossk1* mutants was sensitive to oxidative stressors; the RGI values of Δ*Aossk1* mutant were higher than those of the WT strain on the TG medium supplemented with menadione (0.07 and 0.09 mM) and H_2_O_2_ (5, 10, and 15 mM); for example, the RGI values of Δ*Aossk1* mutant increased by 30–40% after adding 0.09 mM menadione (Appendix A). In contrast, Congo red contributed to mycelial growth; the RGI values of the Δ*Aossk1* mutant were lower than those of the WT strain on the medium supplemented with Congo red (30, 60, and 90 μg/mL). However, SDS had no influence on the mycelial growth in the Δ*Aossk1* mutant (Appendix A).

### 3.8. Aossk1 Regulates Trap Formation, Pathogenicity, and EDs

To further investigate the effects of AoSsk1 on trap formation and pathogenicity, the WT and Δ*Aossk1* mutant strains were cultured at 28 °C for three to four days, after which nematodes were added to observe trap formation and predation. The WT and Δ*Aossk1* mutant strains formed single-loop traps at 12 hpi, and numerous multiple-loop traps at 24, 36, and 48 hpi. There was a remarkable increase in the traps formed by the Δ*Aossk1* mutant compared to those formed by the WT strain (Figure 6A,B). The nematode mortality rate of the Δ*Aossk1* mutants was remarkably higher than that of the WT strain at 12 and 24 hpi, and the nematodes were completely captured at 36 and 48 hpi by the WT and Δ*Aossk1* mutant strains (Figure 6C). In addition, more EDs were observed in the WT strain than the Δ*Aossk1* mutant (Figure 6D).

### 3.9. AoSsk1 Regulates the Secondary Metabolism

After incubation in PD broth for six days, both the hyphae and the supernatants of the WT and Δ*Aossk1* mutant strains were collected and qualified. The dry biomass of the WT hyphae (0.846 g) was 2.15-fold greater than that of the Δ*Aossk1* hyphae (0.392 g). Based on the dry biomass values of hyphae, 200 and 430 mL of the supernatant were collected from the WT and mutant cultures, respectively. The metabolites in the crude extracts of the WT and Δ*Aossk1* mutant strains were detected by LC–MS. Comparison of the chromatograms showed that the WT strain produced abundant metabolites, whereas the levels of metabolites in the Δ*Aossk1* mutant were significantly reduced. The metabolite peak of the Δ*Aossk1* mutant was much shorter than that of the WT strain at 22–36 min (Figure 7A). Heatmap analysis indicated downregulation of many metabolic pathways in the Δ*Aossk1* mutant (Figure 7B). Volcano plot analysis showed considerably more downregulated metabolites in the Δ*Aossk1* mutant than in the WT strain; there were 807 upregulated and 6643 downregulated metabolites in the Δ*Aossk1* mutant (Figure 7C). The differential metabolic pathways between the Δ*Aossk1* mutant and the WT strain were involved in the degradation of the aromatic compounds toluene, naphthalene, and cholesterol; the chorismate metabolism; and the biosynthesis of trichothecene, steroid hormones, gibberellin, scopolin, esculin, and novobiocin (Appendix A). In addition, the metabolites downregulated in the Δ*Aossk1* mutant compared to the WT strain focused on retention time at 22–32 min (Appendix A). NT fungi-specific metabolites (arthrobotrisins) were detected in the Δ*Aossk1* mutant and WT strains (diagnostic fragments ion at *m*/*z* 139, 393, and 429) (Figure 7D and Appendix A), and the relative peak area of arthrobotrisins in the Δ*Aossk1* mutant (MA: 135377790) was less than that in the WT strain (MA: 142665821).

## 4. Discussion

Eukaryotes usually respond to environmental stress by activating the MAPK signaling pathway [41]. The Hog1–MAPK cascade is an important pressure-activated signal transduction pathway that plays important roles in the response to osmotic pressure, oxidative stress, and fungal toxicity [8,9]. Ssk1 is an upstream regulatory protein in the Hog1 signaling pathway, and has been characterized in yeasts and several filamentous fungi [12,13,14,15,16,17]. Herein, we described the characteristics of AoSsk1, an ortholog of *S. cerevisiae* Ssk1, in a representative species of NT fungi. It plays a pleiotropic role in vegetative growth, conidiation, the stress response, and nematode predation in *A. oligospora*.

The deletion of *Aossk1* significantly affected mycelial growth and cell nuclei. The mycelial growth of the Δ*Aossk1* mutant slowed down, the hyphal septum increased, and partial hyphal cells were remarkably enlarged. Meanwhile, the number of cell nuclei in the mycelia and conidia of the Δ*Aossk1* mutant considerably decreased. Similar defects in mycelial growth have been observed in yeasts; the deletion of *ssk1* inhibits pseudohyphal development in *Candida lusitaniae* [42], and Δ*ssk1* null strains show severely reduced hyphal development in *C. albicans* [12]. However, the deletion of *ssk1* inhibits the mycelial growth in *A. alternata* [16], but causes no obvious growth defect in *B. bassiana* [15] or *V. dahliae* [17]. Therefore, these findings suggest that Ssk1 plays varied roles in the mycelial growth in different fungi, including the mycelial growth and cell nucleus development of *A. oligospora*.

Aside from growth defects, the absence of *Aosskl* affected autophagy, LD accumulation, and endocytosis. The number of autophagosomes and LDs was remarkably reduced in the hyphal cells of the Δ*Aossk1* mutant, whereas their volumes were considerably larger than those in the WT strain. Recently, several autophagy-related genes, such as *atg1*, *atg4*, and *atg5*, have been shown to play an indispensable role in conidiation and trap formation in *A. oligospora* [36,43,44]. In addition, the deletion of *Aossk1* inhibited endocytosis and reduced the number of autophagic vacuoles. These novel findings will help probe the relationship between autophagy, LD accumulation, and endocytosis in the hyphal growth and development of *A. oligospora*.

Many filamentous fungi, such as NT fungi, naturally produce asexual conidia for the main reproductive propagule [35]. The deletion of *Aossk1* caused a significant reduction in conidial yield and spore germination rate. Similarly, *Bbssk1* disruption causes a 25% reduction in conidial yield [15]. However, AaSsk1 plays no role in determining the quantity of conidia; the deletion of *Aassk1* in *A. alternata* causes a severe reduction in conidial germination, and also affects the pattern of conidial germination [16]. In addition, the transcription of several sporulation-related genes (*abaA*, *brlA*, *flbC*, *fluG*, and *rodA*) was remarkably downregulated in the Δ*Aossk1* mutant. Of these, *abaA*, *brlA*, and *fluG* are the key regulatory genes involved in conidiation in model and entomopathogenic fungi, such as *A. nidulans* and *A. fumigatus* [45] and *B. bassiana* [46,47]. These findings suggest that Ssk1 plays a crucial role in conidiation in *A. oligospora*, but its role in conidiation varies between different filamentous fungi.

As a response regulator of the Hog1 signaling pathway, Ssk1 is involved in multi-stress responses in yeasts and filamentous fungi. In yeasts, the *C. albicans ssk1* mutant is sensitive to several oxidants, including H_2_O_2_, menadione, and potassium superoxide [13]; moreover, the growth of the *C. lusitaniae ssk1* mutant is slightly affected by high-osmolarity medium, and exhibits hypersensitivity to UV irradiation and high temperature, as well as strong sensitivity to H_2_O_2_ [42]. In filamentous fungi, the *V. dahliae Vdssk1* deletion strains are more sensitive to various stresses, including oxidative stress, but confer higher resistance to fungicides such as fludioxonil and iprodione [17]; similarly, in *B. bassiana*, the *ssk1* deletion strains show significant defects in tolerance to osmotic salts, H_2_O_2_, fungicides, and cell wall biosynthesis inhibitors [15], and the *Aassk1*-deficient mutants show increased sensitivity toward osmotic salts [16]. In this study, the deletion of *Aossk1* caused significant defects in the tolerance to osmotic salts and oxidants. These findings suggest that Ssk1 plays a conserved role and is involved in the response to multiple stresses in fungi.

Previous studies have confirmed that Ssk1 is required for the full virulence of *C. albicans* [12] and several other pathogenic fungi [15,17]. For example, the absence of *Bbssk1* causes a 25% decrease in conidial virulence to second-instar *Spodoptera litura* larvae [15], the *A. alternata ssk1* mutant displays a reduction in necrotic lesions on detached calamondin leaves [16], and the disruption of *Vdssk1* severely attenuates fungal virulence in tobacco seedlings [17]. However, the deletion of *ssk1* (*Brrg-1*) results in no obvious changes in virulence in *Botrytis cinerea* [48]. In this study, the absence of *Aossk1* resulted in increased trap formation and predation efficiency. These findings suggest that Ssk1 is required for fungal virulence, and its ortholog plays a pivotal role in the trap formation and pathogenicity of *A. oligospora*.

NT fungi can produce diverse metabolites during vegetative growth and predator–prey interactions [49,50,51]. Recently, G protein signaling has been shown to be involved in the regulation of the secondary metabolism in *A. oligospora* [23,32]. The deletion of *ric8* (encoding a regulator of G protein) causes a significant downregulation of compounds [23], and the small GTPases Ras2 and Rheb have been found to be necessary for the secondary metabolism in *A. oligospora* [32]. In this study, the loss of *Aossk1* caused a significant reduction in metabolites. Similarly, the disruption of *Vdssk1* results in a significant downregulation of melanin biosynthesis-related genes, as well as reduced melanin accumulation in *V. dahliae* [17]. These findings confirm that AoSsk1 is indispensable for the secondary metabolism in *A. oligospora*.

Although Hog1 and the membrane mucin Msb2 have been characterized in *A. oligospora*, their roles are remarkably different from those of AoSsk1 in terms of phenotypic traits, except sensitivity to high osmolarity. For example, the Δ*hog1* mutant does not exhibit any obvious growth defects under normal growth conditions, and the Δ*msb2* mutant shows no change in conidiation. In particular, the deletion of *hog1* and *msb2* reduces trap formation and predation efficiency [25]. In summary, AoSsk1 plays a multifunctional role by regulating the Hog1 signaling cascade, contributes to mycelial growth and cell nucleus formation, and hinders septum formation. It is also involved in organelle dynamics, such as in that of EDs, LDs, and autophagosomes, and regulates mycelial growth, conidiation, and trap formation. Meanwhile, it affects the production of secondary metabolites. In addition, AoSsk1 can regulate multiple stress responses, including responses to osmotic stress and oxidative stress, and oxidative stress has been closely associated with conidiation and virulence in fungi [44,52]. Therefore, *A. oligospora* can respond to extracellular signals, such as nematodes and various stressors, by AoSsk1 and the Hog1 signaling pathway, and can regulate mycelial growth, trap formation, and lifestyle transitions. Moreover, the deletion of *Aossk1* speeds up trap formation and predation efficiency, so AoSsk1 may have a potential application in the control of nematodes in the future. However, the detailed mechanism by which AoSsk1 regulates phenotypic changes requires further investigation. Herein, we revealed that AoSsk1 is a multifunctional regulator and plays a crucial role in the growth, development, and pathogenicity of *A. oligospora*, and our findings can help probe the roles and regulatory mechanisms of the two-component signaling system in NT fungi.

## 5. Conclusions

Our work showed that AoSsk1 is a conserved regulator involved in multi-stress responses and is essential for conidiation and the secondary metabolism. In addition, AoSsk1 is involved in the regulation of mycelial growth, cell nucleus development, septum formation, and organelle development. Importantly, AoSsk1 also regulates trap formation and thus plays a crucial role in pathogenicity. Our findings provide novel insights into the roles of Ssk1 and the Hog1 signaling pathway in NT fungi, which may help us to develop effective anti-nematode agents.

## Figures and Tables

**Figure 1 jof-08-00260-f001:**
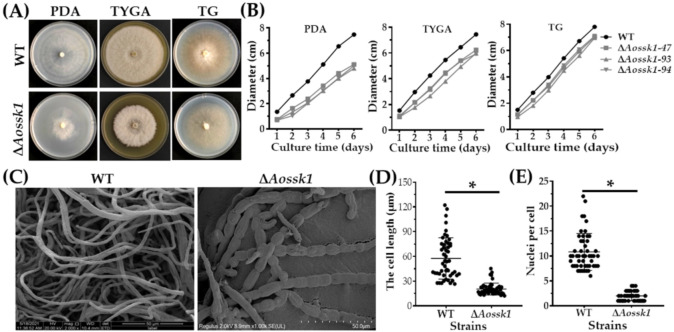
Comparison of mycelial growth, morphology, and cell nuclei between WT and Δ*Aossk1* mutant strains. (**A**) Colony morphologies of WT and Δ*Aossk1* mutant strains cultured on PDA, TG, and TYGA plates for 6 days at 28 °C. (**B**) Colony diameters of WT and Δ*Aossk1* mutant strains cultured on PDA, TYGA, and TG plates. (**C**) Mycelial morphologies of WT and Δ*Aossk1* mutant strains were observed by scanning electron microscopy. (**D**) Comparison of mycelial lengths of WT and Δ*Aossk1* mutant strains. (**E**) Comparison of cell nuclei in the hyphae of WT and *ΔAossk1* mutant strains. An asterisk (**D**,**E**) indicates a significant difference between Δ*Aossk1* mutant and WT strain (n = 50, Tukey’s HSD, *p* < 0.05).

**Figure 2 jof-08-00260-f002:**
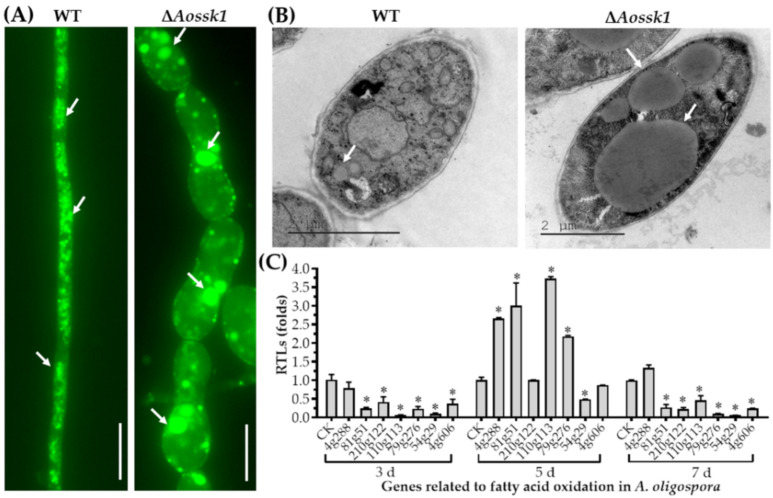
Comparison of lipid droplets (LDs) and transcription of genes related to fatty acid oxidation between Δ*Aossk1* mutant and WT strain. (**A**) Comparison of LDs in the hyphal cells of the WT and Δ*Aossk1* mutant strains. LDs were stained with 10 µg/mL BODIPY dye. Arrows: LDs. Bar = 10 µm. (**B**) Observation of LDs using transmission electron microscopy. Arrows: LDs. (**C**) Comparison of relative transcript levels (RTLs) of genes related to fatty acid oxidation between Δ*Aossk1* mutant and WT strain on the 3rd, 5th, and 7th days. CK (which has an RTL of 1) was used as a standard for the statistical analysis of the RTLs. An asterisk indicates a significant difference between Δ*Aossk1* mutant and WT strain (Tukey’s HSD, *p* < 0.05).

**Figure 3 jof-08-00260-f003:**
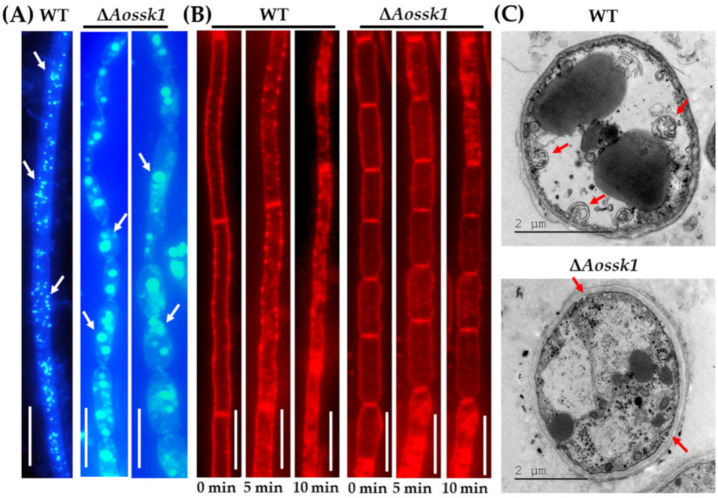
Comparison of autophagy and endocytosis between WT and Δ*Aossk1* mutant strains. (**A**) Comparison of autophagosomes in the hyphal cells of the WT and Δ*Aossk1* mutant strains. White arrows: autophagosomes. Bar = 10 μm. (**B**) Endocytosis in the WT and Δ*Aossk1* mutant strains at different times; Bar = 10 μm. (**C**) Observation of autophagic vacuole in hyphal cells using transmission electron microscopy. Red arrows: autophagic vacuole.

**Figure 4 jof-08-00260-f004:**
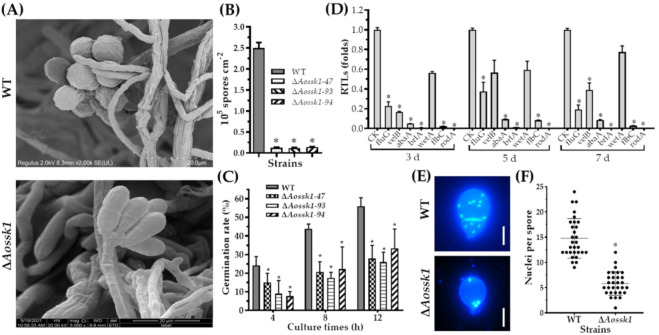
Comparison of conidiation and transcript levels in sporulation-related genes between WT and Δ*Aossk1* mutant strains. (**A**) Spore morphologies of WT and Δ*Aossk1* mutant strains were observed by scanning electron microscopy. (**B**) Conidial yields of WT and Δ*Aossk1* mutant strains. (**C**) Conidial germination rate of WT and Δ*Aossk1* mutant strains. (**D**) Comparison of relative transcript levels (RTLs) of sporulation-related genes between Δ*Aossk1* mutant and WT strain on the 3rd, 5th, and 7th days. CK (which has an RTL of 1) was used as a standard for the statistical analysis of the RTL. An asterisk (**B**–**D**) indicates a significant difference between Δ*Aossk1* mutant and WT strain (Tukey’s HSD, *p* < 0.05). (**E**) Observation of cell nuclei in the conidia of WT and Δ*Aossk1* mutant strains. Bar = 10 μm. (**F**) Comparison of cell nuclei in the conidia of WT and Δ*Aossk1* mutant strains. An asterisk indicates a significant difference between Δ*Aossk1* mutant and the WT strain (n = 30, Tukey’s HSD, *p* < 0.05).

**Figure 5 jof-08-00260-f005:**
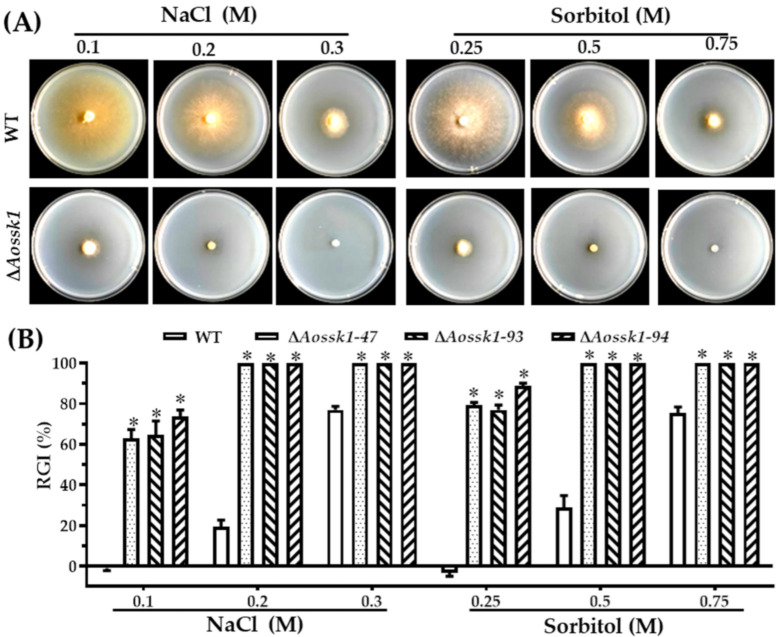
Comparison of osmotic stress responses between WT and Δ*Aossk1* mutant strains. (**A**) Colonial morphology of fungal strains under osmotic stress. (**B**) Relative growth inhibition (RGI) of fungal colonies after being grown for 6 days at 28 °C on TG plates supplemented with different concentrations of NaCl and sorbitol. An asterisk indicates a significant difference between Δ*Aossk1* mutant and the WT strain (Tukey’s HSD, *p* < 0.05).

**Figure 6 jof-08-00260-f006:**
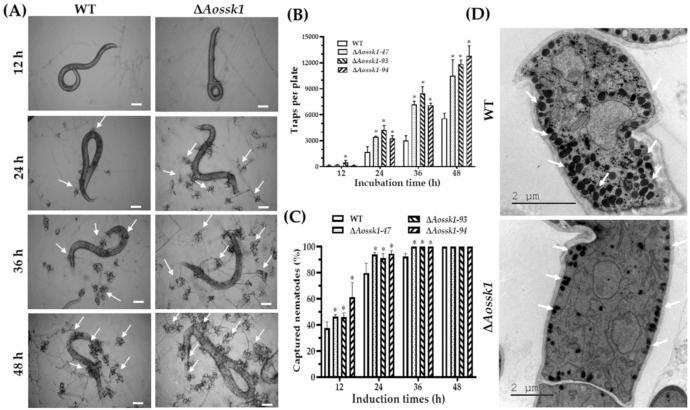
Comparison of trap formation, nematicidal activity, and electron-dense bodies between WT and Δ*Aossk1* mutant strains. (**A**) Trap formation and nematode predation at different time points. White arrows: traps. Bar = 50 μm. (**B**) Comparison of traps produced by WT and Δ*Aossk1* mutant strains at 12, 24, 36, and 48 h. (**C**) Comparison of nematodes captured by WT and Δ*Aossk1* mutant strains at 12, 24, 36, and 48 h. An asterisk (**B**,**C**) indicates a significant difference between the Δ*Aossk1* mutant strain and the WT strain (Tukey’s HSD, *p* < 0.05). (**D**) Observation of electron-dense bodies in trap cells of WT and Δ*Aossk1* mutant strains by transmission electron microscopy. White arrows: ED bodies.

**Figure 7 jof-08-00260-f007:**
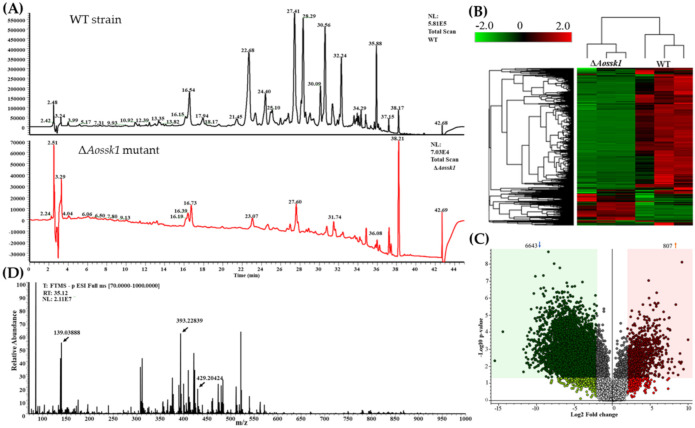
Comparison of metabolic profiling between WT and Δ*Aossk1* mutant strains. (**A**) Comparison of high-performance liquid chromatography profiles of the WT and Δ*Aossk1* mutant strains. (**B**) Heatmap of upregulated and downregulated metabolic pathways between Δ*Aossk1* mutant and WT strain determined via KEGG enrichment. (**C**) Volcano plot of differential metabolites between Δ*Aossk1* mutant and WT strain. (**D**) Mass spectrogram of arthrobotrisins in the WT strain (diagnostic fragments ion at *m*/*z* 139, 393, and 429). RT = 35.12 min.

## Data Availability

Not applicable.

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
