# Peer review of "AoSsk1, a Response Regulator Required for Mycelial Growth and Development, Stress Responses, Trap Formation, and the Secondary Metabolism in Arthrobotrys oligospora"

_jof, 2022, doi:10.3390/jof8030260_

Round 1

Reviewer 1 Report

I do find this work interesting and valuable. The manuscript is well written. Some suggestions (mostly editorial) for improving are below:

Line 104 – S. cerevisiae should be in italics

Lines 236 – 237 – You wrote  “An asterisk (D and E) indicates a significant difference between ΔAossk1 mutant and WT strain (n=50, Tukey’s HSD, p < 0.05)” but I can not find in the figures D and E any asterisks.

Line 287 – you wrote that „the ΔAossk1 mutants only produced 1.26 × 10 5 spores cm -2 (Figure 4B)” However, from Figure 4B it looks like this value is much lower – below 0.2. Please check.

Lines 468 – 469 – here you wrote that”… Meanwhile, it promotes the production of secondary metabolites….”. Please check again this statement. I mean, according to your results there were 807 upregulated and 6643 downregulated metabolites in the ΔAossk1 mutant.

Could you also add in the discussion section what practical applications may have the obtained results?

Author Response

Thank you for your positive comments. We are very appreciative of your great help to improve our manuscript. We have carefully considered and responded to your comments, and list them one by one subsequently.

Line 104 – S. cerevisiae should be in italics

Response: Done. S. cerevisiae has been typeset in italics.

Lines 236 – 237 – You wrote “An asterisk (D and E) indicates a significant difference between ΔAossk1 mutant and WT strain (n=50, Tukey’s HSD, p < 0.05)” but I can not find in the figures D and E any asterisks.

Response: Thank you. We have added the asterisk in the Figures 1D and 1E.

Line 287 – you wrote that „the ΔAossk1 mutants only produced 1.26 × 10 5 spores cm -2 (Figure 4B)” However, from Figure 4B it looks like this value is much lower – below 0.2. Please check.

Response: Thank you. 1.26 × 105 has been changed to 1.26 × 104.

Lines 468 – 469 – here you wrote that”… Meanwhile, it promotes the production of secondary metabolites….”. Please check again this statement. I mean, according to your results there were 807 upregulated and 6643 downregulated metabolites in the ΔAossk1 mutant.

Response: Thank you for your kind suggestion. “it promotes the production of secondary metabolites….” has been changed to “it affects the production of secondary metabolites….”

Could you also add in the discussion section what practical applications may have the obtained results?

Response: Thank you for your kind suggestion. We have added related contents in the discussion section as your suggestion.

Reviewer 2 Report

The authors have made great efforts to characterize an ortholog of Ssk1 (AoSsk1) by gene disruption, multi-phenotypic analysis, and metabolomic approaches. The obtained results provided important information to study the molecular mechanism underlying trap formation and lifestyle transitions of the nematode-trapping fungus. The attained results are of interest to readers and micrologists. Generally, the manuscript has been well documented and organized. However, some weak points have been found and should be revised and edited to improve the quality of the paper. You can find some useful comments and suggestions below:

  1. In the abstract, the authors narrated “Aossk1 disruption caused 95% reduction in conidial yield and remarkable defects in tolerance to osmotic and oxidative stress”.  However, where did this result come from in the text?.
  2.  Introduction part, the authors should highlight the value of the probe the role of the two-component signaling system and Hog1 signaling in NT fungi (further see comments in pdf file).
  3. In Materials and Methods, 

- The growth conditions of A. oligospora (ATCC24927) and the derived mutant strains should be further stated;

- P3, line 99-100: Please briefly indicate how Aossk1 was disrupted

- P3, line 101-102: Please add one sentence that indicates why the authors need to amplify hygromycin fragment hph gene

- P3, line 106: Please add the cited reference for using primer Aossk1-5f/Aossk1-3r

- P3, line 112: Please describe the process to produce ΔAossk1 mutant strains

- P3, line 135-139: Why did the author use those different concentrations? may further discuss in the discussion part

- P4, line 158-159: Please explain why the authors used 2 ×104 spores of the WT and ΔAossk1 mutant strains and about 400 nematodes for analysis of trap formation and nematode predation.

-P4, line 167-168: It should add the reagents for isolation and reverse transcribed to cDNA.P4, line 173: It should add the 2-ΔΔCt method in a brief (further see the comments in pdf file).

Result part:

Fig 1B, if possible it should add the standard error bar, for reliable results

Discussion:

  • The sentences “Similar defects in mycelial growth have been observed in yeasts; deletion of ssk1 inhibited pseudohyphal development in Candida lusitaniae” and  “However, the role of SSk1vary in pathogenic filamentous fungi: deletion of ssk1 inhibited mycelial growth in A. alternata” should be reworded for better clarity and readability.
  • The authors should further discuss the mechanism underlying trap formation and lifestyle transitions of the nematode-trapping fungus;
  • Conclusion should add one sentence to narrate the perspectives of this study
  • English language needs to recheck the spelling, grammar and typing errors
  • Recheck all the references following the style of the submitted journal

Author Response

Thank you for your positive comments. We are very appreciative of your great help to improve our manuscript. We have carefully considered and responded to your comments, and list them one by one subsequently.

In the abstract, the authors narrated “Aossk1 disruption caused 95% reduction in conidial yield and remarkable defects in tolerance to osmotic and oxidative stress”.  However, where did this result come from in the text?.

Response: Thank you for your kind comment.

1) “Aossk1 disruption caused 95% reduction in conidial yield” comes from the result section 3.6 and Figure 4B. The WT strain produced 2.49 × 105 spores cm-2 after culturing on CMY medium for 14 days, the ΔAossk1 mutants only produced 1.26 × 104 spores cm-2, so the conidial yield of the ΔAossk1 mutants was decreased by 95% compared with the WT strain.

2) “Aossk1 disruption caused remarkable defects in tolerance to osmotic and oxidative stress” comes from the result section 3.7, Figure 5, and Figure S4.

 Introduction part, the authors should highlight the value of the probe the role of the two-component signaling system and Hog1 signaling in NT fungi (further see comments in pdf file).

Response: Thank you for your kind suggestion. We have added one sentence to highlight the value of the probe the role of the two-component signaling system and Hog 1 signaing in NT fungi.

In Materials and Methods,

- The growth conditions of A. oligospora (ATCC24927) and the derived mutant strains should be further stated;

Response: Done. The growth conditions of A. oligospora (ATCC24927) and the derived mutant strains have been further stated.

- P3, line 99-100: Please briefly indicate how Aossk1 was disrupted

Response: Thank you for your suggestion. We have described the method of disruption of Aossk1 in this paragraph (2.3).

- P3, line 101-102: Please add one sentence that indicates why the authors need to amplify hygromycin fragment hph gene

Response: Thank you. We have added one sentence that indicates the reason of amplify hygromycin fragment hph gene.

- P3, line 106: Please add the cited reference for using primer Aossk1-5f/Aossk1-3r

Response: Primer Aossk1-5f/Aossk1-3r are designed in this study and listed in Table S2.

- P3, line 112: Please describe the process to produce ΔAossk1 mutant strains

Response: The process to produce ΔAossk1 mutant strains has been described in result 3.2.

- P3, line 135-139: Why did the author use those different concentrations? may further discuss in the discussion part

Response: To compare comprehensively the stress response of the WT and mutant strain to stressors, different concentrations of chemical stressors were used in this study and other studies.

- P4, line 158-159: Please explain why the authors used 2 ×104 spores of the WT and ΔAossk1 mutant strains and about 400 nematodes for analysis of trap formation and nematode predation.

Response: According to our previous study, 2 ×104 spores and about 400 nematodes were used in this study. 2 ×104 spores is feasible to the mycelia growth on the WA plate, and about 400 nematodes can effectively induce trap formation, too much or less nematodes disturb the trap formation and observation.

-P4, line 167-168: It should add the reagents for isolation and reverse transcribed to cDNA.

Response: Thank you. We have added the reagents for isolation and reverse transcribed to cDNA.

P4, line 173: It should add the 2-ΔΔCt method in a brief (further see the comments in pdf file).

Response: We have added the 2-ΔΔCt method in a brief in Method section.

Result part:

Fig 1B, if possible it should add the standard error bar, for reliable results

Response: Thank you. In Fig 1B, the colony diameters of WT and three independent ΔAossk1 mutant strains were compared, we have not added the standard error bar.

Discussion:

The sentences “Similar defects in mycelial growth have been observed in yeasts; deletion of ssk1 inhibited pseudohyphal development in Candida lusitaniae” and “However, the role of SSk1 vary in pathogenic filamentous fungi: deletion of ssk1 inhibited mycelial growth in A. alternata” should be reworded for better clarity and readability.

Response: Thank you for your suggestion. We have deleted the sentence “the role of SSk1 vary in pathogenic filamentous fungi:”.

The authors should further discuss the mechanism underlying trap formation and lifestyle transitions of the nematode-trapping fungus;

Response: Thank you. We have discussed the mechanism underlying trap formation in the last paragraph in Discussion section.

Conclusion should add one sentence to narrate the perspectives of this study

Response: We have added one sentence to narrate the perspectives of this study.

English language needs to recheck the spelling, grammar and typing errors

Response: The revised version has been modified by professional editing service (MDPI English Editing) for improved English writing.

Recheck all the references following the style of the submitted journal

Response: Thank you. We have rechecked all the references.

Reviewer 3 Report

In this study, authors did study AoSsk1, a Response Regulator Required for Mycelial Growth and Development, Stress Response, Trap Formation, and Secondary Metabolism in Arthrobotrys oligospora. In this study, an ortholog of Ssk1 (AoSsk1) was characterized in the nematode-trapping fungus Arthrobotrys oligospora using gene disruption and multi-phenotypic comparison. Deletion of Aossk1 resulted in defective growth, deformed and swollen hyphal cells, increased hyphal septum, and shrunken nucleus. Compared with the wild-type (WT) strain, the number of autophagosomes and lipid droplets in hyphal cells of the ΔAossk1 mutant decreased, whereas their volumes considerably increased. Aossk1 disruption caused 95% reduction in conidial yield, and remarkable defects in tolerance to osmotic and oxidative stress. Meanwhile, the transcript levels of several sporulation-related genes were significantly decreased in the ΔAossk1 mutant compared to the WT strain, including abaA, brlA, flbC, fluG, and rodA. Moreover, loss of Aossk1 resulted in a remarkable increase in trap formation and predation efficiency. In addition, many metabolites were markedly downregulated in the ΔAossk1 mutant compared to the WT strain.

There is nothing new in the study and has a loophole.

  1. The mutant is already made in another study, here the authors have just analyzed it.
  2. The author needs to do experiments with two allelic mutants. It's scientifically wrong to conclude the result based on only just one mutant. either make Ox line or at least one more allelic mutant.
  3. Author another study which was published in "Frontiers in Cellular and Infection Microbiology ' (https://doi.org/10.3389/fcimb.2021.824407 which have less IF than the current Journal) last month, where authors have used two allelic lines which is the right way to do it.

Author Response

Thank you for your comments. But we cannot approve your idea.

The mutant is already made in another study, here the authors have just analyzed it.

Response: We do not find the mutant in another study as your mentioned.

The author needs to do experiments with two allelic mutants. It's scientifically wrong to conclude the result based on only just one mutant. either make Ox line or at least one more allelic mutant.

Response: Thank you for your suggestion. In this study, we used the asexual type of A. oligospora, which only contains a single copy of Aossk1 gene. In addition, three independent mutant strains of Aossk1 gene were generated, respectively, and the corresponding mutant strains were confirmed using PCR method and Southern blotting analysis. These independent mutants for Aossk1 gene showed similar phenotypic properties. Therefore, the phenotypic traits of mutants are reliable and repeated.

Author another study which was published in "Frontiers in Cellular and Infection Microbiology ' (https://doi.org/10.3389/fcimb.2021.824407 which have less IF than the current Journal) last month, where authors have used two allelic lines which is the right way to do it.

Response: Thank you. But we cannot approve your idea.

Round 2

Reviewer 2 Report

The authors have made significant revisions and edits. The paper is fine now. However, some minor formats should be edited such as the names of genes should be in italic, for example, line 642 

Reviewer 3 Report

I am still not satisfied with the author's comments. Two allelic mutants are a must to verify any gene function in science. There are hardly any improvements in the manuscript. Therefore, this manuscript can't be accepted.